# Continual Learning on a Diet: Learning from Sparsely Labeled Streams Under Constrained Computation

**Wenxuan Zhang**[1]**, Youssef Mohamed**[1]**, Bernard Ghanem**[1]**, Philip H.S. Torr**[2]
**Adel Bibi**[2]*__, **Mohamed Elhoseiny**[1]*

[1]King Abdullah University of Science and Technology

[2] University of Oxford

{wenxuan.zhang,youssef.mohamed,bernard.ghanem}@kaust.edu.sa
{philip.torr,adel.bibi}@eng.ox.ac.uk,mohamed.elhoseiny@kaust.edu.sa

## ABSTRACT

We propose and study a realistic Continual Learning (CL) setting where learning algorithms are granted a restricted computational budget per time step while training. We apply this setting to large-scale semi-supervised Continual Learning scenarios with sparse label rate. Previous proficient CL methods perform very poorly in this challenging setting. *Overfitting* to the sparse labeled data and *insufficient computational budget* are the two main culprits for such a poor performance. Our new setting encourages learning methods to effectively and efficiently utilize the unlabeled data during training. To that end, we propose a simple but highly effective baseline, DietCL, which utilizes both unlabeled and labeled data jointly. DietCL meticulously allocates computational budget for both types of data. We validate our baseline, at scale, on several datasets, e.g., CLOC, ImageNet10K, and CGLM, under constraint budget setup. DietCL outperforms, by a large margin, all existing supervised CL algorithms as well as more recent continual semi-supervised methods. Our extensive analysis and ablations demonstrate that DietCL is stable under a full spectrum of label sparsity, computational budget and various other ablations. Our code is available here: https://github.com/wx-zhang/continual-learning-on-a-diet

## 1 INTRODUCTION

In the era of abundant information, data is not revealed in its entirety but rather sequentially from a non-stationary environment. For example, social media platforms, such as YouTube, TikTok, Snapchat, and Facebook, receive huge amounts of data every day. The content of the data and its distribution depend greatly on social trends and focuses on existing platforms (e.g., Facebook, Snap, Twitter), thus shift over time. For instance, Snapchat, in 2017, reported the influx of over 3.5 billion short videos daily from users across the globe (Snap). These videos had to be instantly processed for various tasks, from image rating and recommendation to hate speech and misinformation detection.

Continual learning attempts to address such challenges, focusing on designing training algorithms that accommodate new data streams while preserving previously acquired knowledge. Diverse solutions have emerged, spanning from regularization-based (Kirkpatrick et al., 2017), architecture-based (Ebrahimi et al., 2020), to memory-based methods (Chaudhry et al., 2019b).

Nevertheless, the huge scale of data in most practical applications needs to be processed in real time. Such constraint imposes budget limitations on the continual learning algorithms. To better demonstrate these constraints, imagine if a learning algorithm takes 10 days to learn from a dataset of 3.5 billion samples (accumulated in a single day for Snapchat). The ongoing data stream would have generated 35 billion new samples during the training period. This reality renders the model severely outdated by the time it's put to practical use. Such time constraints on processing pose limitations not only on the number of labels but also on the computation time for the algorithm.

---

*Equal Advising

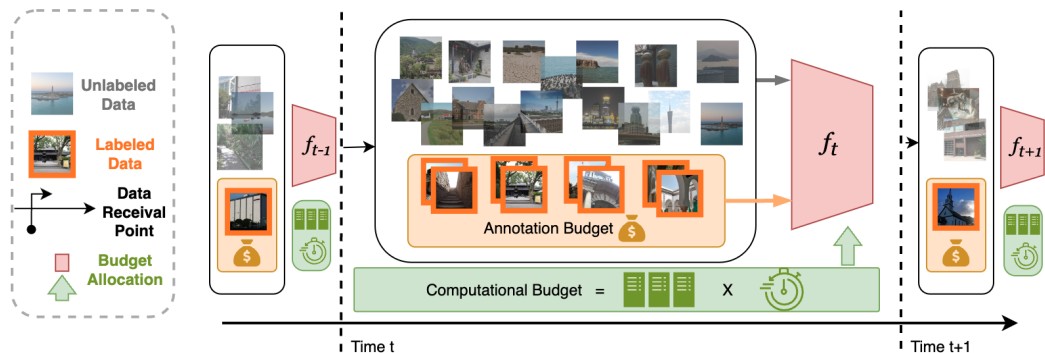

Figure 1: DietCL considers the computation budget due to effective computational time restrictions and very sparse label rate due to annotation cost. At each time step, we propose to allocate sufficient computation for labeled data and utilize the diverse unlabeled data with remaining computation to migrate the overfitting.

Existing literature has recognized the problem and put effort into finding solutions. While some online continual learning approaches attempt to formulate the online data stream (Chaudhry et al., 2019a; Cai et al., 2021), they prioritize the formulation in batch-wise training and evaluation, and often overlook the regularization of computation and time in these algorithms. Additionally, the majority of online continual learning works assume that a full set of labels is accessible. On the other hand, certain advancements in offline continual learning have been achieved by incorporating unlabeled data (Pham et al., 2021; Fini et al., 2022). These approaches often require a full pass of the unlabeled data, thereby neglecting the expensive computational demands associated with large amounts of unlabeled data. Only recently, some work started to consider budgeted continual learning (e.g., (Prabhu et al., 2023)), which aims to regularize the computational requirements for each task to enable the applicability of continual learning algorithms under the aforementioned real-world scenarios. Nevertheless, these endeavors still concentrate on a fully labeled data stream.

In this work, we extend the budgeted continual learning to a semi-supervised manner. This scenario is marked by constrained computational resources and sparse labeling; hence we term it "CL on Diet". The crux of the challenge is illustrated in Figure 1, where substantial data volumes are unveiled during each time step; however, only a fraction of this data is accompanied by labels. Subsequently, the learning algorithms train on these data under the constraint of time use of computational resources, contingent on the time interval of data reception.

Under CL on Diet, we first study the capacity of existing methods to cope with such a challenging setting. Indeed, our findings reveal that current solely supervised methods tend to overfit to the limited labels available. On the other hand, approaches utilizing unlabeled data require extensive computational resources to perform multiple full passes over the vast unlabeled dataset, leading to a significant degradation in performance when computational restrictions are imposed.

To address these challenges, we present our effective baseline, named DietCL. It incorporates a computation budget allocation mechanism to harmonize the learning of current and prior distributions. Moreover, we introduce a unified training strategy that assimilates insights from labeled and unlabeled data concurrently. The efficacy of our baseline is substantiated through evaluations on prominent large-scale continual learning datasets, namely ImageNet10k, CLOC, and CGLM. Our results showcase state-of-the-art performance in this realistic scenario. Additionally, we demonstrate the robustness of our baseline across varying stream lengths, computational constraints, and label rates. Our contributions can be summarized in three folds:

1. We proposed a challenging large-scale semi-supervised continual learning setting, dubbed "CL on Diet", under sparse label scenario and constrained computation budget. We explore problems of the existing methods in this setting.

2. We propose DietCL, a continual learning baseline that utilizes budget efficiently with joint optimization of labeled and unlabeled data.

3. We conduct extensive experiments in large-scale datasets, ImageNet10k, CLOC, and CGLM in data and class incremental settings. Our experiments demonstrate that our simple solution, DietCL, outperforms both supervised and semi-supervised continual learning

algorithms in sparsely labeled streams by 2% to 4%. We also show superior this baseline in varying levels of stream length, label ratio, computational budget.

## 2 RELATED WORK

**Semi-supervised Continual Learning.** Following the recent success of self-supervised learning for pretraining, an increasing body of work is now studying their application for continual learning. Caccia & Pineau showed that self-supervised loss outperformed supervised loss in the meta-learning stage for continual learning. Fini et al. (2022) performed self-supervised pretraining for each task in offline continual learning. Gomez-Villa et al. (2022); Pham et al. (2021); Boschini et al. (2022) used unlabeled data for distillation or regularization loss to migrate forgetting. Most approaches validate their methods on small datasets with a moderate labeled-unlabeled split. They do not consider the computational expense incurred, and often treat labeled and unlabeled data points equally in computation allocation. When we scale up these approaches to cope with the vast amount of real-world unlabeled data and sparse labels, as we shall show in section 3.2, we observe difficulties in learning meaningful label-related information due to their limited computation allocated to labeled data. Additionally, these approaches struggle to learn from unlabeled features due to the constraints of total available computation.

**Scenarios and Budget Constraint in Continual Learning.** Conventional continual learning basically focuses on task incremental learning, class incremental learning, and domain incremental learning. Due to the diversity of the real-world data flow, there has been a lot of work exploring how to eliminate specific constraints in continual learning (Wang et al., 2024), thus making them applicable. Some work considers releasing the task boundary constraint and explores algorithms with unknown boundaries (Aljundi et al., 2019) or blurry task boundaries (Koh et al., 2021). Others consider cases where data is on the fly, and tasks arrive as a one-pass data stream (Chaudhry et al., 2019a). Most of these considerations start from the format of data arrival. In this paper, however, we consider constraints originating from the relation between the processing agent and the data flow.

Only from recent, a question was posed regarding the time limit for training in continual learning. It is demonstrated that if each task can be trained for an unlimited amount of time, a non-continual algorithm can achieve comparable results with continual learning algorithms (Prabhu et al., 2020; Ghunaim et al., 2023). While some online continual learning methods (Koh et al., 2021) report performance based on a fixed number of updates, this is mainly for fair comparisons and not fully explore the impact of the training budget on the algorithm. Recent work (Prabhu et al., 2023) has demonstrated the effectiveness of offline continual learning under limited budgets, showing that learning from a balanced distribution is helpful when budgets are insufficient. Motivated by this, we propose to impose constraints on the training time for semi-supervised continual learning. However, in our work, the budgeted setup is in conjunction with the sparse labeled setting of the stream, which poses a novel more challenging but realistic problem.

## 3 CONTINUAL LEARNING ON A DIET

### 3.1 PROBLEM FORMULATION

In the semi-supervised continual learning under budget, we seek to learn a function $f_\theta : \mathcal{X} \to \mathcal{Y}$ parameterized by $\theta$ that maps images $x \in \mathcal{X}$ to class labels $y \in \mathcal{Y}$. At each time step $t$, the stream samples $n^t$ images $\{x_i^t\}_{i=1}^{n^t} \sim \mathcal{X}^t$ and then only reveals the $n_l^t$ labels of them to $f_\theta$. In contrast to prior work, continual learning algorithms seek to update the parameters $\theta$ with a per time step *predefined computational budget* such that $f_\theta$ performs well on all seen distributions. Throughout this paper, we define the computational budget in terms of the total FLOPs normalized in terms of the number of forward-backward passes, *i.e.*, the training iterations for a given batch size. The computation budget amounts to all FLOP iterations the training requires, from forward-backward pass updating model parameters to any other operations, *e.g.*, importance weights as in Aljundi et al. (2018). We follow Prabhu et al. (2023) for the storage assumption in budgeted continual learning, where the buffer is large enough to store all labeled data while the amount of data used every time is constrained according to the computational budget.

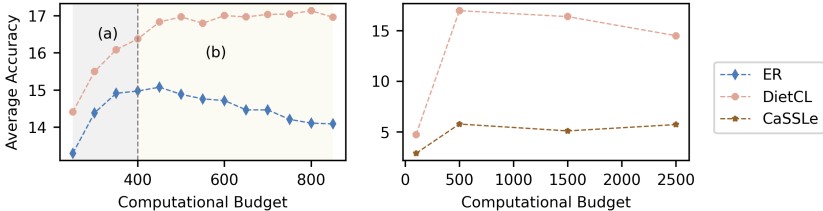

Figure 2: Average accuracy of ER, CaSSLe, and DietCL on 1% labeled ImageNet10k with varying computational steps. Left: supervised method, ER, starts to overfit after 400 steps. Right: semi-supervised method, CaSSLe, converges slowly. DietCL converges fast and alleviates overfitting.

## 3.2 OPPORTUNITIES FOR IMPROVEMENT

Most prior supervised and semi-supervised continual learning works presume sufficient computational resources and labels as they need. Nevertheless, such assumptions may not always be satisfied in some real-world scenarios, including the diet scenario that we propose in this paper. In this section, we explore the potential bottleneck of existing supervised method, ER, and semi-supervised method, CaSSLe, in diet scenario with experiments of varying computational budget per time step on 20-split ImageNet10k, as shown in Figure 2. We compare their performances against our proposed algorithm, DietCL, which shall be introduced in Section 4. Each point of the figure shows the average accuracy at the end of a continual learning stream following Chaudhry et al. (2019a), given the corresponding per time step budget.

**How does Supervised CL behave under a Low Budget?** As shown in region (a) in Figure 2 left where the per step computational budget is less than 400, ER has large performance degradation as the budget goes down. This is partially due to the stability gap (Lange et al., 2023) in ER; during the learning of the new task, the model overfits to the new data first and then recovers knowledge from replayed data of previous tasks. See more details of this phenomenon in Appendix C.1. In the low-budget scenario, the training for the next time step could start before the finish of previous knowledge recovery. Furthermore, the limited number of available labels can also lead to the model capturing a narrow distribution of the current task. Our semi-supervised continual learning method can effectively eliminate overfitting and stability gap issues with the unlabeled data as a regularizer, resulting in improved continual learning performance.

**Can Supervised CL Fully Utilize the Available Computational Budget?** Supervised CL not only struggles to learn efficiently under a limited budget, but suffers overfitting, particularly with a large computational budget, as illustrated in region (b) in Figure 2 when the budget exceeds 400. However, the behavior is different in the semi-supervised method, where overfitting does not happen as much. This encourages us to effectively spend the redundant computational resources on unlabeled data.

**Is Unlabeled Data Necessary?** We demonstrate that even when the computational budget does not reach the maximum capacity for a supervised learning algorithm, *e.g.*, in Figure 2 left, region (a) when the budget is less than 400, we can achieve better results by leveraging unlabeled data to improve generalization when labeled data is limited. Additionally, when the budget is far from enough, *e.g.*, as in Figure 2 left region (b) when the budget is bigger than 400, a purely supervised continual learning algorithm may not fully utilize the computational budget. In this scenario, allocating extra budget to unsupervised data can help capture the current distribution.

**Challenges Facing CaSSLe.** Learning from unlabeled data can be computationally expensive, CaSSLe, as an instance, originally proposed to train for 500 epochs per task. Our experiments reveal how CaSSLe greatly suffers with low budgets in large-scale datasets. As shown in Figure 2 right, the accuracy of CaSSLe remains very low even when we increase the budget to 2500 steps (around 3 epochs). In contrast, the required budget for DietCL to converge is only about 500 steps.

## 4 PROPOSED SOLUTION: DIETCL

We now present our approach that learns from labeled data and unlabeled data jointly to efficiently use budget and capture the shifting distribution. Throughout, we assume that we learn a model $f_\theta$ which can be decomposed to feature encoder $f_{\theta_e} : \mathcal{X} \rightarrow \mathcal{Z}$, where $\mathcal{Z}$ is the feature space, and a

classification head $f_{\theta_c} : \mathcal{Z} \to \mathcal{Y}$, i.e., $f_\theta = f_{\theta_c} \circ f_{\theta_e}$. The learning of each task is constrained with total budget $B = B_u + B_l$, where $B_u$ is the budget for unlabeled data and $B_l$ is the budget for labeled data.

**Learn In Distribution Relationship.** In a large-scale data stream, each time step corresponds to a diverse distribution, which requires a significant effort to learn the in-distribution relationships. To achieve this, we take use of unlabeled data from the current distribution by self-supervised learning (SSL). We discuss how to integrate SSL and which type of SSL to integrate in CL now.

SSL pre-training can be expensive in CL, as observed in Figure 2 (right). Upon further examination in Appendix C.2, we found that the features learned purely from unsupervised loss evolve extremely slowly and are not oriented towards the labels. To address this, we jointly learn from the current unlabeled and labeled data, where the dense unlabeled data acts as a regularizer to prevent overfitting to the sparse labeled data. In terms of the specific SSL algorithms, contrastive learning and masked modeling are two mainstreams. Contrastive learning algorithms typically necessitate augmenting input images into dual views and updating gradients for two distinct backbones. Consequently, with a budget of $B_u$ for unlabeled data, only $B_u/2$ gradient steps are feasible, leading to a significant budget underutilization. As such, we employ an efficient masked modeling method, MAE (He et al., 2022), to capture the current distribution by reconstructing the masked patch of input image. We add a reconstruction head to the encoder, denoted as $f_{\theta_r} : \mathcal{Z} \to \mathcal{X}$, which maps the encoded features to the image space. At each time step $t$, we use the unlabeled stream samples $\{x_i^t\}_{i=n^l+1}^n$ to compute the reconstruction loss as

$$\mathcal{L}_{\mathrm{r}} = \sum_{\forall x_u \in \{x_i^t\}_{i=n^l+1}^n} \sum_{p \in I_p} \left\| f_{\theta_r} \circ f_{\theta_e}(\psi_p(x_u)) - \psi_p(x_u) \right\|^2, \tag{1}$$

where the operator $\psi_p$ extracts the $p^{\mathrm{th}}$ patch from the set of masked patches $I_p$ from every $x_u$.

Furthermore, as proposed by Caccia et al. (2022) that continual learning benefits from learning the current distribution in isolation, we mask out the logits of classes that are not shown in the current time step and compute the following loss function on the labeled data from the current distribution:

$$\mathcal{L}_{\mathrm{m}} = \frac{1}{b_l} \sum_{\{(x_i^t, y_i^t)\}_{i=1}^{n^l}} \mathrm{CE}(f_{\theta_c} \circ I^t(f_{\theta_e}(x_i^t)), y_i^t), \tag{2}$$

where $I^t$ indicate a mask function that zeros out all indices of classes that are not introduced in the current time step $t$ and here CE indicates a cross entropy loss.

To overcome the forgetting problem, we further maintain a task balanced buffer $\mathcal{M}$ that contains only labeled data from the current and previous time steps. The loss can be expressed as follows:

$$\mathcal{L}_b = \frac{1}{b_m} \sum_{(x_i, y_i) \sim \mathcal{M}} \mathrm{CE}(f_\theta(x_i), y_i) , \tag{3}$$

To that end, our final joint loss function to train under budgeted sparsely labeled stream is:

$$\mathcal{L} = \alpha_r \mathcal{L}_{\mathrm{r}} + \mathcal{L}_{\mathrm{m}} + \mathcal{L}_{\mathrm{b}}. \tag{4}$$

Here $\alpha_r$ is the scaling factor to adjusting the mean square loss $\mathcal{L}$ to match the scale of the entropy loss $\mathcal{L}_{\mathrm{m}}$. In the experiment, we fix it to 50.0.

**Budget allocation.** In our final loss function Equation 4, the model is trained on the unlabeled data from the current time step ($\mathcal{L}_r$), labeled data from the current time step ($\mathcal{L}_m$), and labeled data from the balanced buffer ($\mathcal{L}_b$), all jointly. As shown in Figure 2, classic methods have two stages as the budget goes up, i.e., learning stage in the region (a) and overfitting stage in the region (b). The main reason for the overfitting with the second stage is the overuse of the very sparse labeled data of the current task. Therefore, we allocated different training budgets to each data source according to the total budget. We divide it equally among the labeled ($\mathcal{L}_m$), unlabeled ($\mathcal{L}_r$), and buffer data ($\mathcal{L}_b$) if the budget is less than a threshold $B$. Our algorithm converges quickly in this budget, and extending the training duration does not yield significant improvement for current classes. Consequently, we opt to invest the extra budget solely on the buffer data with Equation 3 as our loss function. This stage mainly balances the learning of current and previous classes; refer to Appendix C.3 for detailed analyses. In practice, we choose the threshold $B$ by cross validation; more details are in Appendix B.3. Please refer to Appendix A for the overall training procedure and semantic code.

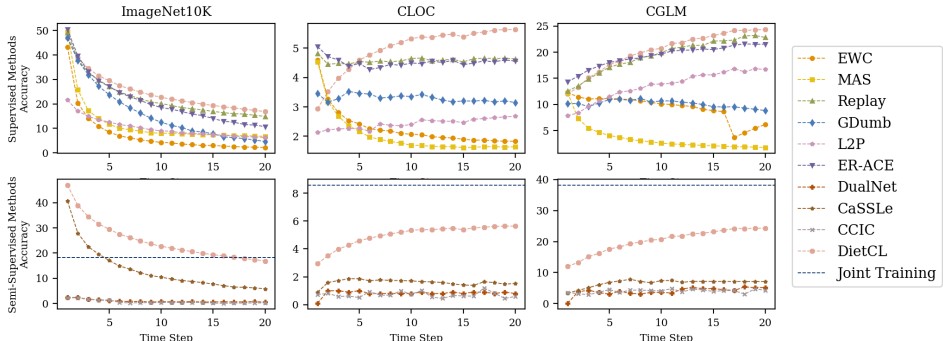

Figure 3: Accuracy at each time step of the baselines on ImageNet10k, CLOC, and CGLM dataset. Our algorithm surpasses the supervised methods by using unlabeled data and outperforms semi-supervised methods due to effective allocation of budgets.

## 5 EXPERIMENTS

In this section, we conduct experiments on various large-scale datasets with our proposed setting, CL on Diet, that the stream is partially labeled and algorithms are granted a limited computational budget per time step. We start with explaining the experimental setup, then we present the comparison of our proposed DietCL against several other methods. At last, we demonstrate that our proposed approach is robust under varying label rate, computational budget and stream length.

### 5.1 EXPERIMENT SETUP

**Benchmark and Metrics.** We use three large-scale continual learning datasets, ImageNet10k, CLOC, and CGLM following Prabhu et al. (2023) to evaluate the performance of DietCL along with other methods. We split ImageNet10k into 20 class-incremental tasks and split CLOC and CGLM into 20 tasks according to the image uploading time shown in the meta information of each image. All the detailed statistics of the datasets can be found in Appendix B.1.

We report the accuracy in the last time step and the average accuracy across the stream. That is to say, let $a_t = \frac{1}{t} \sum_{k=1}^{t} \frac{1}{n} \sum_i \mathbb{1}_{\{f_\theta(x_k^i)=y_k^i\}}$, where $n$ is the number of samples per time step, be the accuracy of the model trained after time $t$ on the valid data until time $t$. Then, we report the accuracy of the union test sets in the last time step $T$ as $\mathcal{A}(T) = a_T$ following Chaudhry et al. (2019a); Fini et al. (2022); Wang et al. (2022). We further report the averaged performance $\mathcal{A} = \frac{1}{T} \sum_{t=1}^{T} a_t$ to show the trend along the whole stream, following (Douillard et al., 2022).

**Training.** Throughout all experiments, we use the ViT model (Dosovitskiy et al., 2020) for classification and the MAE decoder (He et al., 2022) for reconstruction. Both models are pre-trained on the ImageNet1k dataset, released by (He et al., 2022). We set the base learning rate to $10^{-4}$ with a base batch size of 256, and scale the learning rate linearly according to the effective batch size. We use multiple Nvidia A100 GPUs for training, with batch size 256 on each device. We accumulate the loss and perform the backward step when the accumulated batch size reaches 1024. All other learning parameters are adopted from (He et al., 2022). .

**Baselines.** We compare against supervised and semi-supervised continual learning methods in our setting. Supervised methods include ER (Chaudhry et al., 2019b; Prabhu et al., 2023), EWC (Kirkpatrick et al., 2017), MAS (Aljundi et al., 2018), GDumb (Prabhu et al., 2020), L2P (Wang et al., 2022). Semi-supervised methods include CaSSLe (Fini et al., 2022), DualNet (Pham et al., 2021), and CCIC (Boschini et al., 2022). Implementation details of all these methods are left for Appendix B.2. We use balanced sampling for ER; discussions of sampling is in Appendix B.2. We also compare against the joint training, where pre-training and fine-tuning are performed on the whole dataset, following (Hu et al., 2021) to show the upper bound. The per-time-step computational budget for all baselines is the same.

Table 1: Accuracy at the last step of the continual learning sequence. DietCL shows superior performance over the supervised methods when evaluating both the last model and the whole sequence.

| | ImageNet10k | | CLOC | | CGLM | |
|---|---|---|---|---|---|---|
| | $\mathcal{A}(T)$ | $\mathcal{A}$ | $\mathcal{A}(T)$ | $\mathcal{A}$ | $\mathcal{A}(T)$ | $\mathcal{A}$ |
| Upper Bound | 18.23 | - | 8.56 | - | 38.15 | - |
| EWC (Kirkpatrick et al., 2017) | 2.19 | 7.69 | 1.83 | 2.28 | 6.16 | 9.28 |
| MAS (Aljundi et al., 2018) | 6.73 | 11.81 | 1.65 | 2.03 | 1.73 | 3.47 |
| Replay (Chaudhry et al., 2019b) | 14.89 | 22.84 | 4.64 | 4.59 | 22.81 | 19.40 |
| GDumb (Prabhu et al., 2020) | 4.64 | 16.25 | 3.15 | 3.3 | 8.81 | 10.11 |
| ER-ACE (Caccia et al., 2022) | 10.72 | 21.28 | 4.57 | 4.53 | 21.55 | 19.28 |
| L2P (Wang et al., 2022) | 6.30 | 10.23 | 2.68 | 2.43 | 16.69 | 13.57 |
| DualNet (Pham et al., 2021) | 0.50 | 0.92 | 0.80 | 0.83 | 5.20 | 3.97 |
| CaSSLe (Fini et al., 2022) | 5.78 | 13.25 | 1.52 | 1.60 | 7.10 | 6.67 |
| CCIC (Boschini et al., 2022) | 0.20 | 0.62 | 1.32 | 0.72 | 4.27 | 4.02 |
| **DietCL** | **16.82** | **24.90** | **5.63** | **4.98** | **24.34** | **20.26** |

## 5.2 MAIN RESULTS

We conduct experiments on 20-split ImageNet10k with 1% label rate and 500 computational steps, 20-split CLOC with 0.5% label rate and 1000 computational steps, and 20-split CGLM with 5% label rate and 600 computational steps for main comparison. The results of DietCL and the baselines described in Section 5.1 are shown in Figure 3 and Table 1. The figure shows the evaluation accuracy $\mathcal{A}(t)$ at each step, and the table shows the accuracy of the last step $\mathcal{A}(T)$ and the averaged performance $\mathcal{A}$.

**DietCL against Supervised CL.** The first row in Figure 3 shows the comparison between DietCL and supervised CL methods. DietCL consistently outperforms all the supervised methods we compared across all datasets, namely ImageNet10K, CLOC, and CGLM. The most competitive method among the supervised ones is Replay, which involves jointly utilizing current labeled data and previously labeled data uniformly sampled from memory for supervised training. However, the incorporation of unlabeled data in DietCL significantly enhances performance, resulting in accuracy of 16.82%, 5.98%, and 24.34%, on the respective datasets, as shown in Table 1.

This demonstrates that DietCL can effectively leverage the unlabeled data even under the restricted per step computation towards improving the learning in the stream.

**DietCL against Semi-Supervised CL.** The second row in Figure 3 shows the comparison of DietCL against recent semi-supervised continual learning methods and SSL upper bounds. These semi-supervised methods were originally proposed without computational constraint, and encounter a large performance drop with fixed computational budget. Among them, as shown in Table 1, the best performing, CaSSLe, achieves only 5.78% Average Accuracy at the last task on ImageNet10k dataset compared to ours of 16.82%. CaSSLe originally proposed to perform 500 epochs of unlabeled training on their evaluated benchmarks, which roughly equals 50,000 steps in our setting. Our limited 500-step budget causes their performance to collapse. Furthermore, when evaluating semi-supervised baselines in our large-scale dataset, we find some methods, that rely on the inter-class relationship, such as DualNet and CCIC, struggle to cope with such a large number of classes. At the same time, as shown in Figure 3, our fine performance on the three datasets is much closer to the joint training performance than other semi-supervised methods.

**Learning trend.** In the class-incremental benchmark of ImageNet10k, which has around 10K classes, the model is required to learn about 500 new classes at each time step. This poses a significant challenge for most methods to remember the previous massive classes while learning this large amount of new classes. While in the time-incremental benchmark of CGLM and CLOC, the data volume and content of each class vary over time. This requires the model to learn the essence of the category rather than overfitting to the most recent training distribution to overcome the inference bias in the recognition of previous tasks. The overall learning trend of various methods can be seen from the slope of the curves in Figure 3 and the $\mathcal{A}$ in Table 1. Notably, certain methods depicted in the Figure, such as EWC, MAS, and CaSSLe, experience a significant drop in accuracy in the first several tasks, while DietCL avoids large performance declines and gradually enhances accuracy in the CLOC and CGLM datasets. In the table, our method still maintains the highest $\mathcal{A}$ on all three

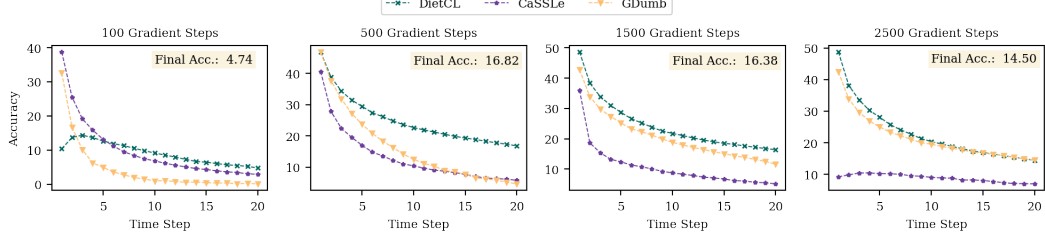

Figure 4: **Varying the Computation per Time Step.** Accuracy of DietCL, CaSSLe, and GDumb with different computational steps at each time step in 1% ImageNet10k. The top right boxes show the average accuracies of DietCL in corresponding computational step settings.

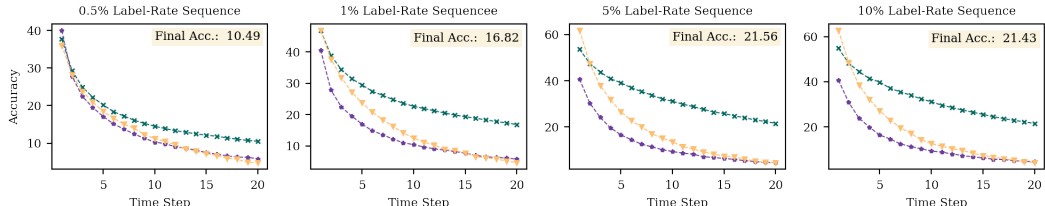

Figure 5: **Varying the Label Rate per Time Step.** Accuracy of DietCL, CaSSLe, and GDumb in data streams with different label rates in 20-split ImageNet10k, with 500 computational steps for each experiment. The top right boxes show the average accuracy of DietCL in corresponding label settings.

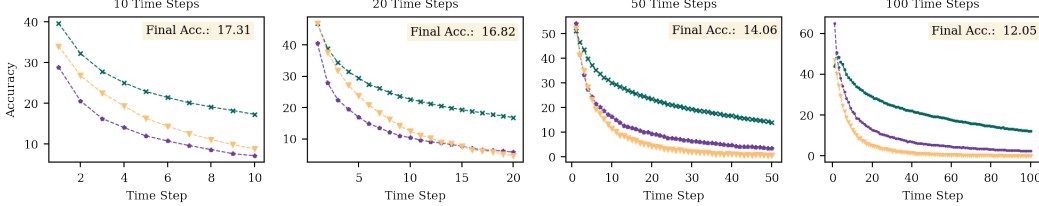

Figure 6: **Varying the Number of Time Steps.** Accuracy of DietCL, CaSSLe, and GDumb with different number of time steps in 1% labeled ImageNet10k. We keep the total computational budget the same for each experiment. The top right boxes show the average accuracy of DietCL in corresponding computational time step settings.

datasets, which are 24.9%, 4.98%, and 20.26%. We believe this is due to the proper allocation of budgets on current and previous data, which allows our method to efficiently leverage the available resources and adapt to the new classes.

Taking the Snap example we mentioned in the introduction, we further connect the results to the real-world problem in Appendix D.

## 5.3 ABLATING EQUATION 4

We conduct ablation study of Equation 4 under 500 computational steps in 1% labeled 20-split ImageNet10k benchmark in Table 2. We start with a Replay baseline, and show the effect by adding the masked classification loss, the balanced buffer, and the reconstruction loss. By adding masked classification loss, the final accuracy increased by 0.9%. By maintaining the balanced buffer, the final accuracy increased by 0.13%. With the help of unlabeled data, the final accuracy increased by 0.86%. This demonstrates that the learning of current distribution

| Replay | $\mathcal{L}_{\mathrm{m}}$ | Balanced Buffer | $\mathcal{L}_{\mathrm{r}}$ | Avg. Acc. |
|:---:|:---:|:---:|:---:|:---:|
| ✓ |  |  |  | 14.97 |
| ✓ | ✓ |  |  | 15.83 |
| ✓ | ✓ | ✓ |  | 15.96 |
| ✓ | ✓ | ✓ | ✓ | 16.82 |

Table 2: Ablation study of our loss function Equation 4. We use 1% labeled stream with 500 computational steps on ImageNet10k.

in isolation, both with masked loss and with unlabeled data, contributes most to the scenario with a limited training budget. We perform ablation with other orders in Appendix B.4

## 5.4 COMPUTATIONAL BUDGET AND LABEL RATE

We study the stability DietCL along with the most robust supervised method GDumb and the best performed semi-supervised method CaSSLe on ImageNet10k against a varying level of label rate, computational budgets, and stream length.

**Varying the Computational Budget.** We conduct experiments with varying the computational budget (100, 500, 1500, 2500) iterations per time step. Across all following experiments, we keep all other parameters unchanged as before, *i.e.*, a label rate of $1\%$ for 20 time steps. The results are summarized in Figure 4.

When the budget is reduced to 100 iterations per step, we can clearly observe a significant decline in performance across all methods. This effect is most pronounced in the case of the supervised method, GDumb, highlighting the minimum resource requirements for achieving satisfactory performance in supervised learning is relatively large. However, it's worth noting that DietCL outperforms the other two methods even under this tight budget constraint, and maintains a stable performance across various budgets, including the scenario of a per-time-step budget of 2500 iterations. Although 2500 iterations might be considered large for DietCL to converge, it remains a constrained setup, particularly in comparison to semi-supervised learning approaches like CaSSLe. As previously mentioned, CaSSLe initially proposed a training regimen involving 50,000 steps, which is 20 times larger than the maximum number of steps we examined. Consequently, expecting CaSSLe to complete its training within just 5% of its original budget is clearly an impractical expectation.

**Varying Label Rate.** Figure 5 shows experiments with various levels of sparsity of the of labeled samples in the data stream, *i.e.*, 0.5%, 1%, 5%, 10%. We keep the computational budgets to 500 steps per time step over 20 time steps. We observe that when raising the label rate from 0.5% to 1%, and from 1% to 5%, we can see obvious increment of the average accuracy in our method, although the computational budget remains the same. However, the performance of GDumb and CaSSLe do not improve a lot. Additionally, the performance gaps between our methods and others are also becoming large, which indicates that our use of labeled data is more efficiently thanks to unlabeled data. When we involve more labeled samples from 5% to 10% in our method, where the budgets are insufficient for the later stream, we do not observe the performance improvement. We conclude that when the budget is sufficient, the labeled data can serve as a better guide for the unsupervised training, and thus allows for improved performance.

**Varying the Number of Time Steps.** We examine the role of the length of the stream on ImageNet10k with experiments on the spectrum of 10, 20, 50, and 100 time steps. This mimics the speed at which data is presented. We use label rate $1\%$ for these experiments. Among the experiments with different stream length, we keep the total number of computational iterations identical to previous experiments by proportionally reducing or increasing the per time step budgets accordingly, so as the total iterations are equal to $20 \times 500$. The results are shown in Figure 6. With the same number of passes of the labeled data, the average accuracies of DietCL of over the sequence are similar, *i.e.*, 24.4%, 24.9%, 23.7%, 22.2%. However, when the task length is longer, both GDumb and CaSSLe have worse performance. This implies that our method is capable of dealing with small-batch high-frequency streams compared to previous methods.

## 6 CONCLUSIONS

We rethink the computational budgets and the sparsity of labels in the real world, which did not get much attention in previous continual learning works. To solve this challenge, we designed an efficient and effective continual learning algorithm, DietCL, that trains the model with labeled and labeled data jointly. We benchmark the setting with large-scale benchmarks ImageNet10k, ImageNet2k, and CGLM. Our method surpasses classic methods and achieves double of the average accuracy of those methods. We also show the superior performance of our method on two other hard benchmarks and analyze the influence of computational budget, the length of the data stream, and the sparsity of the labeled data on the method. We believe DietCL can serve as a starting point towards exploring new CL algorithms under limited computational budget and sparse labels.

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

## A    SEMANTIC CODE OF THE ALGORITHM

See Algorithm 1 for the overall training procedure. Here, during ModifyClassificationHead, we expand the last layer of the classification head according to the total number of current seen classes, and initialize the previous dimensions with previously learned weights. During SplitBudget, we split the total budget to joint training stage and fine-tuning stage.

---

**Algorithm 1** Diet Continual Learning

> **procedure** DIETCL($S, T, \mathcal{D}_{1:T}$)   ▷ Input computational steps per time step $S$, total time step $T$,
> distributions $\mathcal{D}_{1:T}$
> > $\mathcal{M} \leftarrow \{\}$                                                              ▷ Init Buffer
> > **for all** $t \in \{1, \ldots, T\}$ **do**
> > > $\boldsymbol{D}_l, \boldsymbol{D}_u \leftarrow \mathcal{D}_t$                                           ▷ Get offline data
> > > ModifyClassificationHead($\boldsymbol{D}_l$)
> > > $\mathcal{M} \leftarrow \mathcal{M} \cup \boldsymbol{D}_l$                                    ▷ Update Balanced Buffer
> > > $S_1, S_2 = \text{SplitBudget(S)}$
> > > **for all** $s \in \{1, \ldots, S_1\}$ **do**
> > > > $B_l \sim \boldsymbol{D}_l$                                  ▷ Unlabeled batch $B_u$, memory batch $B_m$
> > > > $B_u \sim \boldsymbol{D}_u$                                          ▷ Unlabeled batch $B_u$
> > > > $B_m \sim \mathcal{M}$                                            ▷ Memory batch $B_m$
> > > > $\theta \leftarrow \theta - \nabla \mathcal{L}(B_l, B_u, B_m)$   ▷ Update model parameters by loss function (4)
> > > **end for**
> > > **for all** $s \in \{1, \ldots, S_2\}$ **do**                         ▷ Optionally fine-tune stage
> > > > $B_m \sim \mathcal{M}$
> > > > $\theta \leftarrow \theta - \nabla \mathcal{L}(B_m)$   ▷ Update model parameters by loss function (3)
> > > **end for**
> > **end for**
> **end procedure**

---

## B    MORE DETAILS OF EXPERIMENT

### B.1    BENCHMARK STATISTICS

*ImageNet10k: class-incremental.* We created a large-scale, sparsely labeled ImageNet10k benchmark from the ImageNet21k V2 dataset (Ridnik et al., 2021). To eliminate the possibility of bias to ImageNet1k, particularly when using pretrained models, we first remove the classes present in ImageNet 1k (Deng et al., 2009) from ImageNet 21k V2. To that end, the resulting ImageNet10k benchmark contains 9459 classes with a total of 9,822,675 labeled images. We then split the benchmark into T splits representing T time steps of a stream. We randomly select a fixed ratio of each class in each split as separate labeled data. This procedure of CL data set construction is standard in prior work Chaudhry et al. (2019a); Fini et al. (2022). In each of the time steps, since we assume the stream is only partially labeled, we only use $1\%$ of the labels in each time step.

*CLOC: domain-incremental* We evaluate the domain adaptation ability of our method on CLOC (Cai et al., 2021). This benchmark contains 10788 classes and a total of 38m images for the geo-localization tasks. The images are ordered according to the timestamps at when they were taken, mimicking a natural distribution shift. the stream from this dataset is presented over 20 time steps, with per step $0.5\%$ of the labels are available. That is to say, the stream reveals around 1.9m images, only 1000 of which are labeled.

*CGLM: domain-incremental.* We evaluate the domain adaptation ability of our method on CGLM (Prabhu et al., 2023). This benchmark contains 10788 classes and a total of 581,100 of landmarks from Google Map. The images are ordered according to the timestamps at when they were taken, mimicking a natural distribution shift. Similarly, the stream from this dataset is presented over 20 time steps with per step $5\%$ of the labels are available.[1] That is to say, the stream reveals around

---

[1] We increase the label rate in CGLM to 5% since CGLM is a long tail distribution, and 30% of the classes will only have less than 3 images in 1% labeled stream.

30k images, only 600 of which are labeled. In Table 3, we show the statistics of ImageNet10k and ImageNet2k, and compare them with other popular semi-supervised continual learning benchmarks. Both benchmarks have much larger scales and sparser labels than the previous benchmark. In Table 4, we show the statistics of CGLM and compare it with other recent semi-supervised domain incremental benchmarks, where the domain shifts according to the time. CGLM has a much larger number of classes and sparser labels, and thus is much harder than CLEAR10 and CLEAR100.

| Dataset | Split | Classes per split | Labeled data per split | Unlabeled data per split | Inference data per split |
|---|---|---|---|---|---|
| CIFAR100-semi | 10 | 10 | 1.5k | 14k | 1k |
| ImageNet10k | 20 | 473 | 5k | 500k | 12k |
| CLOC | 20 | 712 | 1k | 1.9m | 19k |

Table 3: Statistics of ImageNet10k with $1\%$ label rate, and comparison with previous class-incremental semi-supervised benchmarks.

| Dataset | Split | Classes | Labeled data per split | Unlabeled data per split | Inference data per split |
|---|---|---|---|---|---|
| CLEAR10 | 11 | 10 | 3k | 700k | 0.5k |
| CLEAR100 | 11 | 100 | 10k | 3.6m | 5k |
| CGLM | 20 | 10788 | 0.6k | 30k | 3k |
| CLOC | 20 | 713 | 1k | 1.9m | 19k |

Table 4: Statistics of 20-split CGLM with $5\%$ label rate, 20-split CLOC with 0.5% label rate, and comparison with previous domain-incremental semi-supervised benchmarks.

### B.2 IMPLEMENTATION DETAILS OF BASELINES

- ER (Chaudhry et al., 2019b). The original proposed methods are one-epoch algorithm. We follow Algorithm 2 Reservoir sampling update in Chaudhry et al. (2019b) to sample from and update the replay buffer for the first epoch of every time step in our training. We assume the buffer size is sufficient large that can store all the training samples. For the remaining epochs, we sampled from the buffer without update. Prabhu et al. (2023) highlighted the uniformly sampling in ER. We compared uniformly sampling and balanced sampling in Table 5 and finally chose to perform balanced sampling that samples 50% data points from the buffer in every mini-batch.

- ER-ACE (Caccia et al., 2022). We use the same buffer update strategy as in ER. We adopt the masking strategy and metric learning method for the classification head from the original paper.

- EWC (Kirkpatrick et al., 2017) and MAS (Aljundi et al., 2018). We follow Avalanche library (Lomonaco et al., 2021) to implement offline EWC and MAS. We searched the hyperparameters and presented the best results.

- GDumb (Prabhu et al., 2020). We follow the standard implementation of the original paper. The mask is set to the seen classes up to the current time step.

- L2P (Wang et al., 2022). The original paper used model weights pre-trained on ImageNet 21k and fine-tuned on ImageNet 1k. We load the ImageNet1k pre-trained weights and

| Method | Uniformly Sampling | Balanced Sampling |
|---|---|---|
| $\mathcal{A}(T)$ | 12.73 | 14.97 |

Table 5: Different sampling strategies in ER. Experiments are on 20-split 1% labeled ImageNet10k benchmark with 500 steps computational budget.

| Supervised Budget | 300 | 350 | 400 | 450 |
|---|---|---|---|---|
| Task 2 | 32.44 | 32.7 | **32.91** | 32.75 |
| Task 3 | 29.30 | 29.75 | 29.80 | **30.02** |
| Task 4 | 26.81 | 27.10 | **27.44** | 27.09 |

Table 6: Average Accuracy up to the corresponding task under varying budget. No matter which task is used for selecting budget threshold, the budget threshold is always set around 400-450.

find suddenly performance drops. The authors proposed to train 5 epochs for every time step without replay buffer, and provided limited buffer size cases as well. However, we find that when we modify the model in our setting to have unlimited replay buffer but limited gradient steps, the performance is even better. The results we present is the modified version.

- CaSSLe (Fini et al., 2022). We chose the Barlow Twins SSL baseline to report the results. The original paper proposed to pre-train for 400 epochs and fine-tune for 100 epochs on ImageNet100. We adopt this ratio $4 : 1$ of pretraining and finetuning steps in our settings where the total budget is fixed. In the finetuning stage, we adopt CaSSLe in a class incremental way by training one linear classifier for all the classes seen so far for a fair comparison. The performance drop comes from the class incremental classifier and limited computation.

- DualNet (Pham et al., 2021). Given that the slow net comprises only a few convolutional layers, we solely load the ImageNet 1k pre-trained weight for the fast net. In the original paper's semi-supervised implementation, the author employed a variable sampling approach to determine whether to use the supervised or unsupervised loss for the current batch, based on a comparison with the label rate. This method substantially augmented the diversity of labeled data. In our implementation, we first segregate the current data into labeled and unlabeled subsets and then adopt the variable sampling approach of the original paper to decide whether to sample from the labeled or unlabeled subset. We determine the threshold to be 0.5, which yields the best results, but lower than the original ones. Additionally, we include all the gradient steps, including the two view transformation of the contrastive loss, in the computation and fix the total number of gradient steps, which leads to further performance degradation. Although the original paper proposed task-agnostic and task-free training strategies, we experimented with both but evaluated the model in a class-incremental manner. Neither strategy yielded favorable results in the context of large-scale class-incremental streams with limited computational resources.

- Upper bound (He et al., 2022). In particular, SSL Joint Training trains through self supervision on all the unlabeled data once and then fine-tunes on all the available labeled data. The total computational budget given is the effective budget of continual learners, which is 20 *time steps* × *budget per time step*. 80% of this budget is spent on self supervision, while the other is for fine-tuning.

### B.3 BUDGET THRESHOLD

We chose to conduct cross validation to choose the threshold $B$ to separate the balanced training stage and learning on buffer stage. The cross validation is only on the initial three tasks out of 20 tasks; this is standard and widely accepted in the field of continual learning (Chaudhry et al., 2019a).

To show the robustness of such selection, we performed experiments in ImageNet10k benchmark with different supervised budgets and compared their performance in tasks 2, 3, and 4 in Table 6. We present the average accuracy up to the corresponding task. When choosing the supervised budget from either task 2, 3 or 4, the chosen budget threshold is consistently similarly from 400 to 450.

### B.4 ABLATION WITH OTHER ORDER

Here in ablation study, we show different orders as well as the observations and report the average accuracy of the last model.

Table 7: Ablation study with different orders in 20-split ImageNet10k benchmark

| Replay | $\mathcal{L}_\mathrm{m}$ | Balanced Buffer | $\mathcal{L}_\mathrm{r}$ | Avg. Acc. | Replay | $\mathcal{L}_\mathrm{m}$ | Balanced Buffer | $\mathcal{L}_\mathrm{r}$ | Avg. Acc. |
|---|---|---|---|---|---|---|---|---|---|
| ✓ | | | | 14.97 | ✓ | | | | 14.97 |
| ✓ | | ✓ | | 15.01 | ✓ | | ✓ | | 15.01 |
| ✓ | ✓ | ✓ | | 15.96 | ✓ | ✓ | | ✓ | 15.24 |
| ✓ | ✓ | ✓ | ✓ | 16.82 | ✓ | ✓ | ✓ | ✓ | 16.82 |

The first table shows that the task-balanced buffer can marginally improve the Replay method. Then, both $\mathcal{L}_\mathrm{m}$ and $\mathcal{L}_\mathrm{r}$ can improve the algorithm by around 1%. This is consistent with our original observation.

The second table shows that without masked loss, the unlabeled data can only marginally improve the performance. This further validates the point that the loss of the unlabeled data should be guided by proper supervised loss, as stated in section 4.

## C  ANALYSIS

### C.1  ER AND DIETCL IN LOW COMPUTATIONAL BUDGETS

In Figure 7, we present the validation loss and accuracy of classes originally introduced from time steps 0, 1, and 2, during the training of time step 2 for DietCL and ER when the total computational step is 300. Our results indicate that the classical continual learning algorithm, ER, exhibits the stability gap phenomenon, as proposed by Lange et al. (2023). Specifically, during the learning of new classes, the accuracy of previously learned classes initially drops, before eventually resuming an upward trend once the learning of new classes stabilizes. Notably, in ER, although the accuracy of current classes initially rises to approximately 40%, it subsequently declines significantly at a later time step, before gradually improving again. Moreover, when the budget is fully expended, the learning of previous classes remains unfinished.

However, in our algorithm, the accuracies of previously learned classes, as depicted by the blue curves in the figure, do not exhibit a significant drop, and the learning of new classes is smoother. In other words, the accuracy of current classes does not undergo the "up-and-down detour" observed in ER, which results in significant budget savings. We postulate that in ER, since the labeled data is sparse, the model is more susceptible to overfitting to the space represented by the sparse data from current classes. Conversely, the incorporation of unlabeled data in our algorithm guides the learning towards a more generalized space, thereby eliminating the overfitting issue and the stability gap. As a consequence, our algorithm is efficient in learning under low-budget scenarios, especially when the labeled data is sparse.

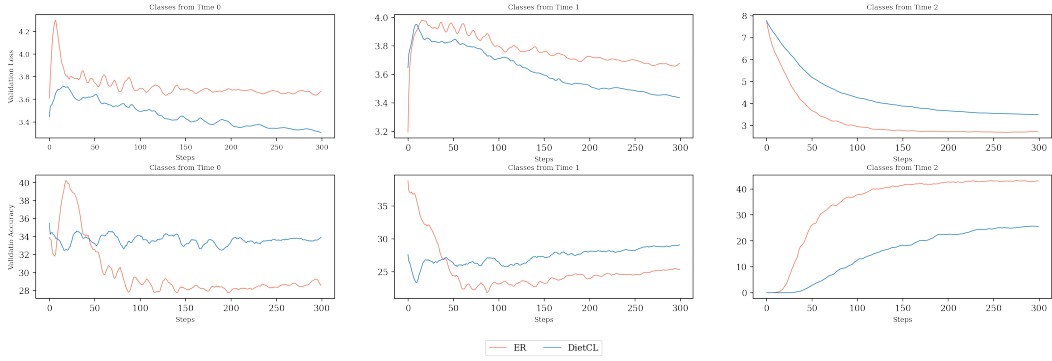

Figure 7: Validation Loss and Accuracy of classes introduced from time step 0,1,2 during the training of time step 2. The training is conducted on ImageNet10k, with label rate 0.01 and a budget of 300 steps.

## C.2 DIRECTLY PERFORM PRE-TRAINING AND FINETUNING IN CL

We conducted an empirical study on the effectiveness of using SSL methods in continual learning. One naive approach is to perform two-stage training iteratively, consisting of pre-training and fine-tuning stages, which leverages unlabeled data in the pre-training stage. To establish baselines, we implemented the `OneStage` and `TwoStage` methods, both of which replay all the labeled data from the current time step. In the `OneStage` method, we randomly sample batches from all seen classes and use cross-entropy loss for classification. In the `TwoStage` method, we first perform MAE pre-training with the unlabeled data of the current time step using a fixed ratio of computational budgets, followed by fine-tuning the model with labeled data randomly sampled from the buffer using the remaining budgets. We perform this experiment in an extremely sparse labeled stream with a sufficient large computational budget. Note that this setting is already highly permissive towards semi-supervised training compared to supervised training scenarios, as the budget is relatively large and the amount of labeled data is extremely sparse.

Our results, as shown in Figure 8, reveal that an extra unlabeled stage training in the `TwoStage` method improves the classification accuracy of the samples from the *new classes*, which are the exact classes from which the unlabeled images used in the pre-training stage are obtained. This observation is consistent with the success of self-supervised learning works (He et al., 2022). However, we did not observe any advantages of the pre-training stage when measuring the performance among *all seen classes*. We hypothesize that the pre-training phase harms the label-related representations learned in previous time steps, resulting in a drop in the Average Accuracy of the `TwoStage` method.

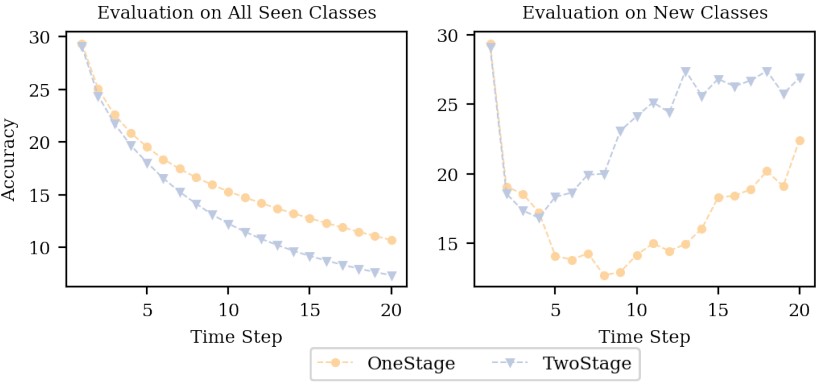

Figure 8: Accuracy of the One Stage MAE and Two MAE baseline on $0.5\%$ labeled streams. Each baseline has 2500 steps computational budget at each time steps.

## C.3 EFFECT OF THE FINE-TUNING STAGE

In Figure 2, we utilize the initial 400 computational steps to perform joint training of the model using labeled and unlabeled data along with data from the balanced buffer. For budgets that exceed 400, the remaining steps are allocated for fine-tuning using solely data from the buffer. The selection of 400 steps was based on the observation that classical methods tend to sacrifice performance beyond this threshold, signifying that it is adequate for learning the current distribution. As an illustration, we consider the experiment involving the 20-split ImageNet10k with a label rate of 0.01 and 600 computational steps, where we compare the training outcomes with and without fine-tuning in Figure 9. The figure demonstrates that the balanced fine-tune applied during the final stages of training leads to more balanced, thus improved, results.

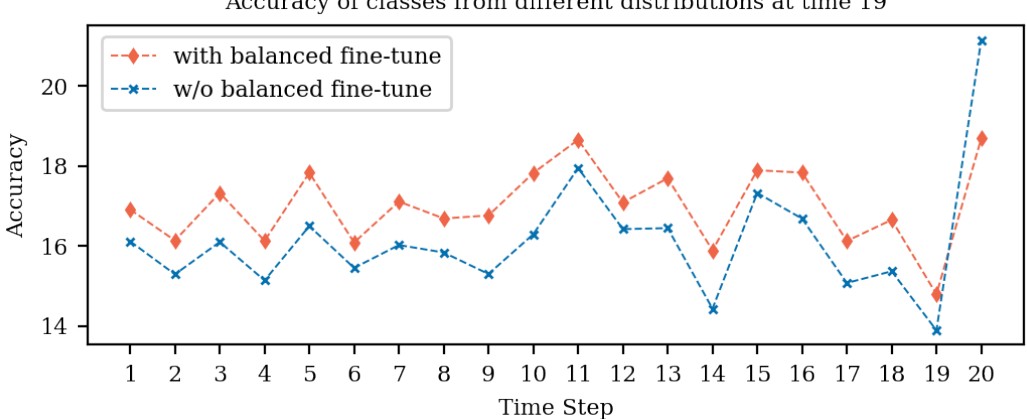

Figure 9: Accuracy of classes from each of the previous distribution after the training of the last time step. After fine-tuning the model on the balanced buffer, the accuracies across the stream become more balanced.

## D  CONNECTION TO REAL-WORLD PROBLEMS

We aim to study a scenario where we cannot conduct a full epoch on all incoming data, but can allocate enough computation for the sparsely labeled data. Our algorithm discusses how to take use of extra budget from labeled data and put it in unlabeled data.

In our experiment, we need (at least) 0.5 GPU hours on an 80G Nvidia A100 to train each task on the 1% labeled 20-split ImageNet10k stream. Each task contains a total of 1m images and 10k labeled images, and we perform 500 gradient steps with a batch size of 1024.

Now, we want now to translate the computational requirement that we used in this paper (as detailed above) to the scale of Snap example. As mentioned in the introduction, Snap receives around 3.5B videos every day. Take the ranking tasks by classifying N snapshots of each video as an example, there will be $3.5N$ billion image data every day. We assume there are $L$ annotators for this task and each can produce 4000 labels per day. To match our 500 computational steps, we expect the budget to be at least $B = 4000L/10000 * 500 = 200L$. As we need 0.5 GPU hours for 500 steps in our experiment, this $200L$ amounts to around $100L$ GPU hours. That is, when $B/L$, the computation budget per labeling labor, is around 100 GPU hours (almost a whole day running on 4 GPUs), our algorithm can outperform other baselines in the pre-defined ranking tasks. And the upper bound for the performance is related to the label rate $4000L/(3.5 * 10^9 N)$.

We note that the time required is closely related to the GPU type, as well as data loading time, GPU communication time, and other factors. For instance, in a system with slow data loading, we may need up to 18 GPU hours. Through collaboration with various experts in acceleration, the $B/L$ (budget per labeling labor for the algorithm to converge) can be further reduced.

