# OpenReview forum: "Continual Learning on a Diet:  Learning from Sparsely Labeled Streams Under Constrained Computation"
_ICLR.cc/2024/Conference — ICLR 2024 poster_

### Official Review · Reviewer_uZcN · 2023-10-27

**Soundness:** 4 excellent
**Presentation:** 4 excellent
**Contribution:** 3 good
**Rating:** 8
**Confidence:** 4

**Summary:**

The authors study continual learning with a restricted computational budget (defined by FLOPs) per time step. Demonstrating that pre-existing supervised and semi-supervised methods struggle in this setting with overfitting or lack of adequate computation resources, the new semi-supervised method DietCL is proposed.

**Strengths:**

- The setting of limited compute and labeled data is important and realistic, and the proposed method is well-adapted to this setting
- Defining compute budget by FLOPs allows us to consider being unable to even complete a full epoch, which is challenging for learning but useful and realistic for large datasets and small compute capabilities
- Evaluation of other prior methods seems to be fair and complete

**Weaknesses:**

- Fig 2 shows that DietCL does still suffer from overfitting, just not as bad as other methods (particularly ER) and takes much larger computational budget for the effect to be bad. In the caption and paper text authors claim DietCL doesn't suffer from overfitting, which seems to be a bit of an overstatement

**Questions:**

- In the task-balanced buffer $\mathcal M$ introduced in Section 4, why do you include just data from the current and previous time step? If budget allows it, why not include data from more time steps or ensure you have an adequate diversity of examples from different classes included in the buffer?
- Questions about the reported empirical numbers: How many runs are these reported across (for instance to produce the figures 2-6) and how large is variance across the runs? Does the ordering examples are presented during training impact learning a lot, and are some methods more robust to this than others?

---

> ### Author Response · Authors · 2023-11-22
> **Response to Reviewer uZcN**
>
> We thank the reviewer uZcN for recognizing the motivation and the effectiveness of our algorithm, as well as the extensive experiments,  and for their valuable suggestions. Please see our response to the feedback below.
>
> **1. DietCL doesn't suffer from overfitting, which seems to be a bit of an overstatement**
>
> Thank you for bringing this out. We’ve modified this to “alleviate overfitting” in blue words in the main paper.
>
> **2. If budget allows it, why not include data from more time steps**
>
> We are sorry about the misleading. The text should be “contains only labeled data from the current and previous time **steps**”. In our approach, the buffer includes data from the current and **all** previous time steps.
>
> **3. The influence of randomness**
>
> Our current result is average over 4 random seeds with mean 16.82125% , std 0.015%.
>
> We also run one experiment with different sequence order of DietCL in the setting we report in the Table 1, i.e., 1% label rate and 20-split-ImageNet 10k stream with  500 computational steps. The comparison of two sequence with the same random seed is as follows
> |  | $\mathcal{A}(T)$ | $\mathcal{A}$ |
> |---|:---:|:---:|
> | Sequence Order 1 | 16.674 | 24.697 |
> | Sequence Order 2 | 16.824 | 24.9 |
>
> We note that there’s around 0.1% difference in different class order. And we use the same class order for all the ablation results reported in the paper.
>
> We would not have time for more runs during this stage, and we will add more comparisons (10 runs) with sequence order and random seed in later version

---

### Official Review · Reviewer_dSE4 · 2023-10-29

**Soundness:** 3 good
**Presentation:** 3 good
**Contribution:** 3 good
**Rating:** 8
**Confidence:** 3

**Summary:**

This paper proposes “CL on Diet”, which is a large-scale semi-supervised continual learning setting, which is an extension of budgeted continual learning to a semi-supervised setting.
DietCL utilizes budget efficiently with joint optimization of labeled and unlabeled data, and comes with a computation budget allocation mechanism to harmonize the learning of current and prior distributions.
The experiments demonstrate that DietCL outperforms both supervised and semi-supervised continual learning
algorithms in sparsely labeled streams by 2% to 4%.

**Strengths:**

1. By asking several questions, Section 3.2 successfully gives the motivation of DietCL by showing the shortcomings of existing methods under low budget scenario.
2. DietCL is well-designed with sufficient details given. For example, using unlabeled data as a regularizer, DietCL can effectively eliminate overfitting and the stability gap issues.
3. Sufficient details are given for the experiment part.

**Weaknesses:**

1. Comparison with (Prabhu et al., 2023) is needed in the experiment section, since it belongs to the supervised continual learning methods.
2. need more connections to real-world problem: It is unclear how far is a real-scenario from the low budget setting given in this paper.  It would be better to further discuss a real example (such as snapchat example), by quantifying how many budget per time step is actually possible for a real scenario (with typical hardware).
3. The Ablation study uses 'Replay -> Lm -> Balanced Buffer -> Lr', how about trying 'Replay -> Lm -> Lr -> Balanced Buffer'  or another order. some different observations may be obtained.
4. small presentation issues
  (1) Confused notation. In Appendix A, subscriptions {l,u,m} are used to demonstrate labeled, unlabeled and buffer data. While in section 4, {r,m,b} are used.
   (2) typo in Figure 6 'to make them easier to compare?'

**Questions:**

see weaknesses

---

> ### Author Response · Authors · 2023-11-22
> **Response to Reviewer dSE4 [1/2]**
>
> We thank the reviewer dSE4 for recognizing the motivation and the effectiveness of our algorithm, as well as the extensive experiments,  and for their valuable suggestions. Please see our response to the feedback below.
>
> **1. Comparison with (Prabhu et al., 2023)**
>
> We appreciate the reviewer’s suggestion to include a comparison with the work presented in Prabhu et al., 2023. They proposed an important baseline, denoted as ER - uniformly sampling, where they augment the buffer with current data and  their training batches are uniformly sampled from all seen task, including the data from the current task and previous task.   In our paper, we also compare with ER in our modification in sampling strategy. We perform balanced sampling of the current task data and buffer data.
>
> We provide a comparison of this baseline with our method here on the ImageNet10k dataset.
> | Method  | $\mathcal{A}(T)$ |
> |---|:---:|
> | ER - uniformly sampling (Prabhu et al., 2023) | 12.73 |
> | ER - balanced sampling | 14.97 |
> | DietCL | **16.82** |
>
> The ER-balanced sampling can improve ER-uniform sampling by around 2%, and our method can further improve ER - balanced sampling by approximately 2%. We argue that the main cause of the different observations of our paper and (Prabhu et al., 2023)  is the class number scale. Prabhu et al. primarily evaluated uniform sampling in 2000 total classes with class incremental learning, whereas we have 10k classes and require more learning in inter-task relations. That's why our ER modification, which involves balanced sampling from current and previous tasks, shows better results.
>
> Due to time limit, we cannot finish our experiment of ER - uniformly sampling on CGLM and CLOC. We will add this in our future version.
>
> **2. Need more connections to real-world problem**
>
> We thank the reviewer for bringing up this valuable point. Before discussing a specific example in the real world, we want to re-clarify our principle in designing the setting.
>
> We aim to study a scenario where we cannot conduct a full epoch on all incoming data, but can allocate enough computation for the sparsely labeled data. This is not highly related to the total data amount, but to the expected budget and the amount of labeled data. In fact, our algorithm discuss how to take use of extra budget from labeled data and put it in unlabeled data. The performance worsens when the budget decreases or the number of labeled data goes down, as shown in Figures 4 and 5.
>
> In our experiment, we need (at least) 0.5 GPU hours on an 80G Nvidia A100 to train each task on the 1% labeled 20-split ImageNet 10k stream. Each task contains a total of 1m images and 10k labeled images, and we perform 500 gradient steps with a batch size of 1024.
>
> Now, we want now to translate the computational requirement that we used in this paper (as detailed above) to the scale of Snap. As mentioned in the first paragraph of the introduction, Snap receives around 3.5B videos every day. If they want to perform ranking tasks by classifying N snapshots of each video, there will be 3.5N billion image data every day. Assume they can label $4000 * L$ images every day (we assume each labor's labeling speed is 4096 per day, and L is the number of labors). To match our 500 computational steps,
> we expect the budget to be at least $B = 4000 * L / 10000 * 500 = 200L$. As we need 0.5 GPU hours for 500 steps in our experiment, this  200L  amounts to around $100L$ GPU hours. That is, when $B/L$, the computation budget per labeling labor, is around 100 GPU hours (almost a whole day running on 4 GPUs), we can train a good ranking system with our algorithm. And the upper bound for the performance  is related to the label rate $4000L / 3.5*10^9N$.
>
> We note that the time required is closely related to the GPU type, as well as data loading time, GPU communication time, and other factors. For instance, in a system with slow data loading, we may need up to 18 GPU hours. Through collaboration with various experts in acceleration, the $B/L$ (budget per labeling labor for the algorithm to converge) can be further reduced.

---

> ### Author Response · Authors · 2023-11-22
> **Response to Reviewer dSE4 [2/2]**
>
> **3. Trying ablation with other order**
>
> Thank you for your suggestion to explore different sequences in our Ablation study. We chose the order 'Replay -> Lm -> Balanced Buffer -> Lr' to validate the effectiveness of learning from labeled data through various approaches like replay, masked loss, and balanced buffer, followed by the addition of Lr as a regularizer to use unlabeled data. Here we show different orders as well as the observations and report the average accuracy of the last model.
>
> The new two ablations are
> | DietCL components | $\mathcal{A}(T)$ |
> |---|:---:|
> | Replay | 14.97 |
> | Replay + balanced Buffer | 15.01 |
> | Replay + balanced buffer + Lm | 15.96 |
> | Replay + balanced buffer + Lm + Lr | 16.82 |
>
> | DietCL components | $\mathcal{A}(T)$ |
> |---|:---:|
> | Replay | 14.97 |
> | Replay + balanced Buffer | 15.01 |
> | Replay + balanced Buffer + Lr | 15.242 |
> | Replay + balanced buffer + Lm + Lr | 16.82 |
>
> The first table shows that the task-balanced buffer can marginally improve the Replay method. Then, both Lm and Lr can improve the algorithm by around 1%. This is consistent with our original observation.
>
> The second table shows that without masked loss, the unlabeled data can only marginally improve the performance. This further validates the point that the loss of the unlabeled data should be guided by proper supervised loss, as stated in Appendix C and section 4.
>
> **4. Presentation issues**
>
> Thank you for bringing to our attention the presentation. We have changed them highlighted in blue color in the rebuttal verrsion.

---

### Official Review · Reviewer_TT9Y · 2023-10-29

**Soundness:** 3 good
**Presentation:** 3 good
**Contribution:** 3 good
**Rating:** 6
**Confidence:** 3

**Summary:**

This paper explores a problem setting of data efficient continual learning, where only a small portion of available data is labeled and the use case demands fast training compute. The paper introduces a continual learning algorithm that combines self-supervised on the unlabeled data with the conventional supervised continual learning strategy on the labeled subset. The algorithm divides the compute budget equally between these components to demonstrate that the proposed method outperforms existing approaches and helps to reduce overfitting.

**Strengths:**

The problem setting is specific and there is detailed discussion on the challenges of existing methods. The proposed method is simple and seems to perform well under many ablations.

**Weaknesses:**

1. The overall method is quite simple, which in itself is fine, but would warrant extensive ablation on the design choices. For instance, the algorithm depends on MAE as a way to generate good pre-trained features. How would the algorithm perform with alternative representation learning strategies? Furthermore, the buffer threshold is determined via cross-validation. How difficult is this cross-validation to perform and how many time steps / tasks are needed to learn it?
2. The ablations provided seem to compare against the semi-supervised algorithms. However, it is clear from Table 1 that the comparable semi-supervised algorithms are outperformed by DietCL, whereas supervised techniques such as ER and ER-ACE are competitive and almost as good as DietCL. It would have been more interesting to see ablations against these supervised learning baselines, for example to demonstrate the diet limits under which the proposed method is challenged.
3. Notation in page 6 defining A(t) and A seem to be incorrect due to overloading $t$

**Questions:**

What is the non-monotonic behavior of DietCL in Fig 4?

---

> ### Author Response · Authors · 2023-11-22
> **Response to Reviewer TT9Y [1/2]**
>
> We thank the reviewer TT9Y for recognizing the research direction and the effectiveness of our algorithm and for their valuable suggestions. Please see our response to the feedback below.
>
> **1. How would the algorithm perform with alternative representation learning strategies?**
>
> The suggestion to explore other representation learning algorithms is indeed valuable. Before delving into alternative representation learning strategies, we wish to emphasize that our choice of MAE was driven not only by its feature extraction capabilities but mostly due to its efficiency.
>
> As revised in blue in Section 4, within the current landscape of SSL (self-supervised learning) methods, most contrastive learning-based approaches require augmenting the input image into two views and processing them at least twice, sometimes necessitating multiple iterations over the input. This practice reduces the number of effective iterations within a given computational budget. We opted for MAE as it presents a more computationally efficient alternative. Although there might be concerns regarding the additional decoder backbone required by MAE, evidence from the original MAE paper [B] and subsequent works like SupMAE[A] suggests that employing a single linear layer for the decoder can yield comparable performance. In this manner, we need only process each image once for MAE loss, maintaining the model architecture almost identical to that used in supervised learning.
>
> While there are other SSL methods that do not demand such augment on input examples. These methods typically involve image masking and reconstruction, like Beit v2 [C]. But it necessitates extra training for their proposed codebook and feature comparison between the teacher model, incurring additional computational costs.
>
> **2. How difficult is this cross-validation to perform and how many time steps / tasks are needed to learn it?**
>
> The cross-validation procedure we employed is straightforward. We chose to conduct it only on the initial three tasks out of 20 tasks; this is standard and widely accepted in the field of continual learning[D].
>
> We performed experiments with different supervised budgets and compared their performance $\mathcal{A}(t)$ in tasks 2, 3, and 4. We present the average accuracy up to the corresponding task. For a varying number of tasks used to choose the buffer threshold, the selected threshold is consistently similar.
>
> | Supervised Budget | 300 | 350 | 400 | 450 |
> |:---|---|---|---|---|
> | Task 2 | 32.44 | 32.7 | **32.91** | 32.75 |
> | Task 3 | 29.30 | 29.75 | 29.80 | **30.02** |
> | Task 4 | 26.81 | 27.10 | **27.44** | 27.09 |
>
>
> **3. Ablations against these supervised learning baselines(ER)**
>
> The analysis reveals that ER demonstrates limited efficacy under a large computation budget, as shown in Figure 2. Meanwhile, GDumb has a higher potential in large budget scenarios since it allocates the budget equally to all task data, as shown in Figure 4 at 2500 steps. As such, we initially perform a computation budget ablation study with GDumb, and follow this in other ablations.
>
> We agree with reviewer that ER is a competitive baseline in low-budget scenarios, as we show in Figure 2, and Table 1. We further include it in our ablation study here. Due to limited resources at this stage, we present ER results under various low-budget conditions (First three columns in Figures 4, 5, and 6 ). We will update with the full results in a later version.
>
> As the computation ablation can be seen from Figure 2, here we only perform label rate ablation and task length ablation.
>
> Label rate ablation (Figure 5):
> | Label rate | 0.005 | 0.01 | 0.05 |
> |---|:---:|:---:|:---:|
> | ER | 7.86 | 14.89 | 19.67 |
> | DietCL | **10.49** | **16.82** | **21.56** |
>
> Task length ablation (Figure 6):
> | Task length | 10 | 20 | 50 |
> |---|:---:|:---:|:---:|
> | ER | 14.93 | 14.89 | 13.76 |
> | DietCL | **17.31** | **16.82** | **14.06** |
>
> From these results, DietCL still remains superior performance over ER. As the labeling rate increases with the provision of additional labels, the disparity between ER and DietCL diminishes. As the task length increases where more learning is required on cross-task relation, performance of DietCL becomes similar to ER.
>
> **4. Notation in page 6 defining A(t) and A seem to be incorrect due to overloading**
>
> We indeed agree, this was a confusing typos that we have revised and fixed in the rebuttal version of the paper.
>
> Specifically, we report $a_t$ the performance of model trained after time step $t$ on valid set of time step $t$,  $\mathcal{A}(T)$ the performance of last model on valid set from all time step, $\mathcal{A}$ the average over all $a_t$, as the averaged performance of the model along the continual learning procedure.

---

> ### Author Response · Authors · 2023-11-22
> **Response to Reviewer TT9Y [2/2]**
>
> **5. What is the non-monotonic behavior of DietCL in Fig 4?**
>
> We thank the reviewer for bringing this valuable point.  For the first several task in the left most figure of figure 4, the budget is very low. Operating within this constraint, DietCL also needs to allocate its budget towards unlabeled data, resulting in comparatively less computational steps for labeled data than what is typical for supervised methods. Consequently, this leads to worse performance on the labaled data withinthe first task.
>
> However, during the second task, with replaying data from the first task and balanced sampling from the two tasks, the learning of the first task is further enhanced during the learning of the second task. In contrast, other supervised methods spend all the budget on labeled data, enabling them to achieve higher initial performance in the first task.
>
> [A] Liang, Feng, Yangguang Li, and Diana Marculescu. "Supmae: Supervised masked autoencoders are efficient vision learners." arXiv preprint arXiv:2205.14540 (2022).\
> [B] He, Kaiming, et al. "Masked autoencoders are scalable vision learners." Proceedings of the IEEE/CVF conference on computer vision and pattern recognition. 2022.\
> [C] Peng, Zhiliang, et al. "Beit v2: Masked image modeling with vector-quantized visual tokenizers." arXiv preprint arXiv:2208.06366 (2022).\
> [D] Chaudhry, Arslan, et al. "Efficient Lifelong Learning with A-GEM." International Conference on Learning Representations. 2018.

---

### Official Review · Reviewer_GSPx · 2023-10-31

**Soundness:** 2 fair
**Presentation:** 2 fair
**Contribution:** 2 fair
**Rating:** 5
**Confidence:** 3

**Summary:**

This paper presents DietCL to conduct semi-supervised continual learning under a label and computation budget. The main idea is to formulate a loss function that considers a reconstruction loss, a masking loss, and a budget loss at the same time.

**Strengths:**

* Continual learning under a constrained labeling and computational budget is an interesting problem.
* The approach seems to perform well in the evaluated scenarios.

**Weaknesses:**

* The paper lacks clarity and is not well-organized. For instance, there is a related work section (Sec. 2), but the discussion of prior methods continues all the way until page 5 and there is a separate section (Sec. 3.2) for additional coverage of prior work. The challenges of prior work is reiterated numerous times in the first 5 pages, and despite their mention in the abstract and introduction, they are mentioned again on page 4-5 “Challenges facing existing semi-supervised continual learning algorithms.” It is not until page 5 that DietCL is introduced and its coverage is limited to less than a page, which leads to an incomplete presentation of the method.
* The method’s novelty and clear exposition is lacking. What is the key insight here? It seems that the method boils down to using an existing self-supervised learning method (MAE) coupled with masking out logits of classes not shown in the current time step.
* The loss function (Eq. 4) simply adds the three different loss terms without any hyper-parameters in front of them to scale them appropriately. For instance $\mathcal L_r$ is the reconstruction loss (Euclidean distance squared of vectors), and it is being added to the loss function with two other losses that are cross entropy. It seems like there will inevitably be a scaling issue. Even if additional hyper-parameters, e.g., $\alpha_r, \alpha_m, \alpha_b$ were added in front of the loss terms, this would entail hyperparameter optimization to find the right values.

**Questions:**

1. What is the motivation for the loss function in (4), where each term is weighted equally? How would this generalize to other settings where equal-weighting may not lead to desired behavior (e.g., it might overemphasize the reconstruction loss over others, or the budget loss over others)?
2. What is the key novelty that enables the method to outperform prior approaches? Is it the explicit consideration of the budget in the loss function?

---

> ### Author Response · Authors · 2023-11-22
> **Response to Reviewer GSPx [1/2]**
>
> We thank the reviewer GSPx for recognizing the research direction and the effectiveness of our algorithm and for their valuable suggestions. Please see our response to the feedback below.
>
> **1. The paper lacks clarity and is not well-organized.**
>
> We thank the reviewer for raising the clarification and structure issue. First we want to re-clarify our motivation and contribution, and then we resolve the concerns one by one in the following.
> - As we mentioned in Figure 1 and the top part of page 2, we can see that nowadays continual learning is rarely adopted in real-life applications, though a lot of papers have come out. We believe this may stem from the fact that some assumptions inherent in continual learning cannot be satisfied in practical scenarios.  This realization led us to introduce a budget constraint in semi-supervised continual learning.
> - In section 3.1, we introduce a setting where labels are sparse and the computational budget is limited. While both situations have been studied separately [A, B, C], the challenge arises from the joint constraint: unlabeled data typically comes in at a large scale, and the corresponding learning algorithms usually converge slowly. This makes it difficult for the algorithms to learn valuable representations when computational resources are limited.
> - In section 3.2, we conducted experiments to identify performance gaps in existing methods under our setup, which further motivates the need for our approach.
> - In Section 4, we designed an efficient algorithm, DietCL, to jointly learn from labeled data with sampling design and unlabeled data with self-supervised loss. In the bottom part of page 5 in Section 4, we discussed how we adapted this algorithm to cope with budget constraints.
>
> **The challenges of prior work is reiterated numerous times in the first 5 pages, and despite their mention in the abstract and introduction.**
>
> we have modified the structure of the paper to make it easier to follow.
> - We renamed Section 3.2 to 'Opportunities for Improvement,' where we primarily discuss the limitations of current continual learning methods in our setup.
> - We trimmed the text down by removing details related to the methods to the related work section.
>
> **There is a separate section (Sec. 3.2) for additional coverage of prior work**
>
> The purpose of Section 3.2 is to provide motivation for the research direction and algorithm design, as we have restated below and highlighted in the paper with blue text.
> - Budget constraint is indeed a research problem worth exploring, one that previous algorithms are unable to effectively address.
> - The bottleneck in applying previous algorithms in this setting is the sparse label and computation budget.
> We want to note that it is uncommon that paper start with motivational set of experiments [J].
>
>
> **2. The method’s novelty and clear exposition is lacking.**
>
> Each of our design choices is informed by a our best understanding of the problem and the limitations inherent in existing methods.
> - As stated in the second and third paragraphs of Section 3.2, we observed that supervised algorithms tend to overfit to the sparse labeled data, and this phenomenon becomes more severe as the budget increases. To address this issue, we allocated some of the budget to unlabeled data and designed a two-phase training approach to utilize the extra budget on buffer data.
> - As stated in section 3.2 ”Challenges facing CaSSLe”, we found that existing semi-supervised algorithms is computational extensive, and that’s why we believe the unsupervised training should be jointly performed with supervised training.
>
> We wish to clarify that our objective is to develop efficient solutions for each of the problems we observed, thereby enhancing the potential real-world applicability of our algorithm. This is similar to that of prior works such as [A, D, E, F].
>
> **3. The loss function (Eq. 4) simply adds the three different loss terms without any hyper-parameters in front of them to scale them appropriately. What is the motivation for the loss function in (4), where each term is weighted equally?**
>
> This is a typo and we thank the reviewer for spotting this. In our experiment, we used a single  scaling factor to bring the MSE loss $\mathcal{L}_r$ to the same scale as the entropy loss $\mathcal{L}_m, \mathcal{L}_b$. We choose this hyperparameter  mainly follow SupMAE [I] and  by cross validation.  This has now been corrected in our manuscript.
>
> For the loss over buffer and current data, we control the number of samples used from each source to balance their importance, instead of using a scaling factor. This is equivalent to the application of a scaling factor.

---

> ### Author Response · Authors · 2023-11-22
> **Response to Reviewer GSPx [2/2]**
>
> **4. What is the key novelty that enables the method to outperform prior approaches?**
>
> We thank the reviewer for raising this question. Our response is outlined below, and we have adjusted our paper to clarify these points.
>
> As discussed in Section 3.2, a key issue with semi-supervised algorithms is the high computational cost due to the slow convergence of self-supervised loss. Our approach outperforms existing methods in two key aspects of efficiency:
> - Firstly, we chose MAE because it eliminates the need to augment input images into dual views and avoids gradient updates for two separate backbones. This significantly improves computational efficiency compared to the SSL loss in [C].
> - Secondly, as we stated in section 4 and appendix C, we treat self-supervised loss as a regularizer rather than a primary optimization target, given its slow convergence. This approach differs from the pre-training and fine-tuning strategy used in [C].
>
> As stated in the first three bolded points in Section 3.2, the key issue for supervised algorithms is the slow convergence due to the stability gap[G] and the tendency to overfit. We introduce unlabeled data to mitigate overfitting to limited labeled data.
>
> We would like to note that, although most of our designs are derived from existing algorithms, none of them individually works as well as ours. While L2P[H] also utilized a mask for the current task, and [A] discussed ER in a budgeted setting, these components individually do not outperform our approach, as shown in table 1. Here, we add a comparison of the sampling in [A], our modified and reported balanced sampling for ER, and DietCL, using the ImageNet 10k dataset.
>
> | Method  | $\mathcal{A}(T)$ |
> |---|:---:|
> | ER - uniformly sampling [A] | 12.73 |
> | ER - balanced sampling  | 14.97 |
> | DietCL | 16.82 |
>
>
> [A] Prabhu, Ameya, et al. "Computationally Budgeted Continual Learning: What Does Matter?." Proceedings of the IEEE/CVF Conference on Computer Vision and Pattern Recognition. 2023.\
> [B] Pham, Quang, Chenghao Liu, and Steven Hoi. "Dualnet: Continual learning, fast and slow." Advances in Neural Information Processing Systems 34 (2021): 16131-16144.\
> [C]Fini, Enrico, et al. "Self-supervised models are continual learners." Proceedings of the IEEE/CVF Conference on Computer Vision and Pattern Recognition. 2022.\
> [D] Prabhu, Ameya, Philip HS Torr, and Puneet K. Dokania. "Gdumb: A simple approach that questions our progress in continual learning." Computer Vision–ECCV 2020: 16th European Conference, Glasgow, UK, August 23–28, 2020, Proceedings, Part II 16. Springer International Publishing, 2020.\
> [E] Janson, Paul, et al. "A Simple Baseline that Questions the Use of Pretrained-Models in Continual Learning." NeurIPS 2022 Workshop on Distribution Shifts: Connecting Methods and Applications. 2022.\
> [F] Panos, Aristeidis, et al. "First Session Adaptation: A Strong Replay-Free Baseline for Class-Incremental Learning." Proceedings of the IEEE/CVF International Conference on Computer Vision (ICCV), 2023, pp. 18820-18830.\
> [G] De Lange, Matthias, Gido M. van de Ven, and Tinne Tuytelaars. "Continual evaluation for lifelong learning: Identifying the stability gap." The Eleventh International Conference on Learning Representations. 2022.\
> [H] Wang, Zifeng, et al. "Learning to prompt for continual learning." Proceedings of the IEEE/CVF Conference on Computer Vision and Pattern Recognition. 2022.\
> [I] Liang, Feng, Yangguang Li, and Diana Marculescu. "Supmae: Supervised masked autoencoders are efficient vision learners." arXiv preprint arXiv:2205.14540 (2022).\
> [J] Wu, Dongxian, Shu-Tao Xia, and Yisen Wang. "Adversarial weight perturbation helps robust generalization." Advances in Neural Information Processing Systems 33 (2020): 2958-2969.

---

### Author Response · Authors · 2023-11-22
**General Response**

We thank the reviewers for their thoughtful and valuable feedback. We are encouraged that reviewers GSPx, TT9Y, and uZcN found our research direction interesting. To understand the challenges posed by such constraints, we examined existing algorithms under a fixed budget with varying budget choices, which motivated our algorithm design. We are glad that reviewers TT9Y, dSE4, and uZcN recognized our intentions in doing this.  We introduced DietCL, an efficient semi-supervised continual learning algorithm designed to operate within a constrained budget, and conducted thorough large scale experiments to validate its effectiveness. We are grateful to reviewers GSPx, TT9Y, and dSE4 for recognizing our superior results compared to existing baselines and our extensive experiments across various scenarios.

Following reviewers’ suggestions, we modified our writing in blue words in the updated version of paper. We conducted several experiments to show the validity for our method in the corresponding response to the reviewers, and will later add full version to the main paper.
- We include baseline of ER + uniformly sampling from (Prabhu et al., 2023).  (Q4 in response to GSPx and Q1 in response to dSE4)
-  We add comparison with ER in ablation study.  (Q3 in response to TT9Y)
- We report more statistics to show the stability of our method with different random seed. (Q3 in response to uZcN)

---

### Meta-Review · Area_Chair_yruM · 2023-12-06

**Metareview:**

The submission proposes an algorithm (DietCL) for use in the Continual Learning setting with sparse labeling budget as well as limited computation budget. The authors demonstrate that DietCL, which uses a combination of unsupervised and supervised loss functions, is able to significantly outperform 8 different baseline methods across three large-scale CL benchmark datasets in the restricted label/compute setting.

Reviewers tended to agree that the problem are is important, the algorithm design is clear, and that the empirical results are convincing. Much of the issues around clarity and organization were addressed in the revision, with a few questions remaining that I expect the authors to address in the final version of the paper (e.g. sensitivity to buffer size and loss balancing term).

**Justification For Why Not Higher Score:**

The paper does a good job of empirically validating a new technique in what reviewers agreed is an important setting. However, beyond empirical performance, there is not much justification or understanding as to why the method works, which would make it a stronger paper.

**Justification For Why Not Lower Score:**

All reviewers agree that the studied setting is important and that the proposed method provides convincing empirical benefits.

---

### Decision · Program_Chairs · 2024-01-16

Accept (poster)